# Synergistic Effects of Plant Derivatives and Conventional Chemotherapeutic Agents: An Update on the Cancer Perspective

**DOI:** 10.3390/medicina55040110

**Published:** 2019-04-17

**Authors:** Raffaele Pezzani, Bahare Salehi, Sara Vitalini, Marcello Iriti, Felipe Andrés Zuñiga, Javad Sharifi-Rad, Miquel Martorell, Natália Martins

**Affiliations:** 1Endocrinology Unit, Department of Medicine, University of Padova, via Ospedale 105, 35128 Padova, Italy; raffaele.pezzani@gmail.com; 2AIROB—Associazione Italiana per la Ricerca Oncologica di Base, 3520128 Padova, Italy; 3Student Research Committee, School of Medicine, Bam University of Medical Sciences, Bam 44340847, Iran; 4Department of Agricultural and Environmental Sciences, Milan State University, via G. Celoria 2, 20133 Milan, Italy; sara.vitalini@unimi.it (S.V.); marcello.iriti@unimi.it (M.I.); 5Department of Clinical Biochemistry and Immunology, Faculty of Pharmacy, University of Concepción, Concepcion 4070386, Chile; fzuniga@udec.cl; 6Food Safety Research Center (Salt), Semnan University of Medical Sciences, Semnan 3519899951, Iran; 7Department of Nutrition and Dietetics, Faculty of Pharmacy, University of Concepción, Concepcion 4070386, Chile; 8Faculty of Medicine, University of Porto, Alameda Prof. Hernâni Monteiro, 4200-319 Porto, Portugal; 9Institute for Research and Innovation in Health (i3S), University of Porto, 4200-135 Porto, Portugal

**Keywords:** synergy, synergistic effects, anticancer, plant-drug interaction, conventional chemotherapeutic agents, plant derivatives

## Abstract

Synergy is a process in which some substances cooperate to reach a combined effect that is greater than the sum of their separate effects. It can be considered a natural “straight” strategy which has evolved by nature to obtain more efficacy at low cost. In this regard, synergistic effects may be observed in the interaction between herbal products and conventional drugs or biochemical compounds. It is important to identify and exploit these interactions since any improvement brought by such kind of process can be advantageously used to treat human disorders. Even in a complex disease such as cancer, positive synergistic plant–drug interactions should be investigated to achieve the best outcomes, including providing a greater benefit to patients or avoiding adverse side effects. This review analyzes and summarizes the current knowledge on the synergistic effects of plant–drug interactions with a focus on anticancer strategies.

## 1. Introduction

Multidrug therapy is a helpful strategy focused on the direct blocking or killing of damaging agents (such as cancer cells or pathogens) as well as on the activation of human body defenses or repair mechanisms. It derives from a gradual withdrawal of the previously adopted dogma of mono-drug therapy; for decades, pharmacological research was based on the identification of a single active principle [1]. As regards phytotherapy research, only recently have the contributions of traditional Chinese medicine, Ayurveda and traditional Western phytomedicines begun to be scientifically confirmed and appreciated. Furthermore, during the last 20 years, the world has encountered an increasing rate of the use of conventional medicines combined with complementary and alternative medicine (CAM), which are represented not only by homeopathy, naturopathy, chiropractic, and energy medicine, among others, but also ethnopharmacology and phytotherapy [2]. It is becoming evident that many diseases have a multifaceted etiology, which could be treated more successfully with a drug combination strategy than a single administration. In Western countries, for multifactorial or complex disease treatment (e.g., cancer, hypertension, metabolic and inflammatory diseases, acquired immune deficiency syndrome (AIDS) and infections), an effective multidrug therapy is commonly adopted [3]. In this context, phytotherapy and ethnopharmacology play a key “natural” role, as they find their effectiveness from herbs or plants, which are “secundum naturam”, a complex pool of millions of molecules. Of note, human pharmacotherapy started with the use of plants in ancient times, probably imitating animal self-medication, while first written documents date back to the third millennium with the Sumerian empire [4].

Because of the enormous popularity of CAM (including ethnopharmacology and phytotherapy), it is necessary to focus our attention on the risk–benefit profile of herbal remedies and updated information. Thus, the primary aim of this short review is to provide an overview of plant–drug interactions, with a special update on the synergistic anticancer effects between herbal medicines and conventional chemotherapeutic drugs, relying on recently published works and systematic reviews following the guidelines of Pautasso [5]. The literary search strategy was done interrogating PubMed, Web of Knowledge, Google Scholar and ScienceDirect databases, selecting the following keywords: “herb (s)”, “plant (s)”, “phyto*”, “drug (s)”, “medicine”, “interaction (s)”, “cancer”, “tumour (tumor)”, and “chemotherapy”. Algae derivatives or extracts were intentionally excluded, as their investigation would require a dedicated review.

## 2. Synergy

Paying attention to the quote from Buckminster Fuller (1968), “*Universe is synergetic. Life is synergetic*” [6], synergy is broadly defined as the interaction or cooperation of two or more substances, organizations or other agents to produce a combined effect greater than the sum of their separate portions [7]. Synergy comes from the Greek word “synergos”, which means “working together.” More precisely, synergism or synergy according to the McGraw–Hill Concise Dictionary of Modern Medicine is indicated as “the cooperative interaction between two or more components of a system, such that the combined effect is greater than the sum of each part”. In anatomy, it is the combined action of muscle groups which results in a force greater than that which could be generated by the individual muscles, while in pharmacology, synergism is described as an approach to multidrug-resistant bacterial infections or virulent malignancies, among others, in which the use of therapeutic agents may affect different pathways, making the treatment more efficient [8]. Besides this, Mosby’s Dictionary of Complementary and Alternative Medicine defines synergism or pharmacological synergy as the effect of combined components interacting to produce new and different effects than individual components, referring typically to the action of whole plants, as opposed to the active constituents in isolation [9]. As the above statements clearly show, the definition of synergy may overlap to potentiation, a term that means a synergistic action, in which the effect of two drugs administered simultaneously is greater than the sum of the effects of each drug given separately [9]. Nevertheless, a concrete definition derives only from a mathematical approach, demonstrated and proven by pioneering works of Berenbaum, that provided the basis for its use in pharmacology and phytopharmacology [10,11,12]. Of note, the isobole method of Berenbaum is not a unique method that emerged from applied sciences; different methodologies have been developed to understand drug–drug interactions, such as the isobologram method of Loewe [13], the fractional product method of Webb [14] and the combination index method of Chou and Talalay [15].

## 3. The Isobole Method

The isobole method is the most convenient and experimentally suitable method for evaluating synergistic effects. This method is represented by a graph in which a curve is constructed by plotting different drug dose combinations; in the x-axis, the dose rates of the first given drug or are is plotted (X-compound dose), while in the y-axis, the dose rates of the second given drug or compound are imposed (Y-compound dose) (Figure 1).

The points derived from the combination of the two dose rates lie on three different isobole curves. Briefly, an isobole is an “iso-effect” curve, in which a combination of compounds is represented on a graph, as follows:(1)No interaction curve: according to Berenbaum, this is an additive curve (straight line), in which the effects of 2 compounds are the simple sum of the single effects, and they do not interact.(2)Synergistic curve: this concave-shaped curve represents the real synergism (or potentiation), where the effects of 2 compounds are more than the simple sum of the single effects when the 2 compounds are given together.(3)Antagonistic curve: this convex shape curve represents the opposite effects of synergism; i.e., the effects of 2 compounds are less than the simple sum of the single effects.

In mathematical terms, three equations may be derived:OE (d_a_, d_b_) = OE (d_a_) + OE (d_b_)(1)
OE (d_a_, d_b_) > OE (d_a_) + OE (d_b_)(2)
OE (d_a_, d_b_) < OE (d_a_) + OE (d_b_)(3)
where OE means the observed effects, while d_a_ and d_b_ are the doses of the X-compound and Y-compound, respectively. Note that the isobole method is independent of the mechanism of action of drugs and may apply to most circumstances. Furthermore, it is possible to obtain a synergistic effect at one specific dose combination and antagonistic effects at another, even with the same compounds, and certainly, this fact may complicate the isobole curve with a wave-like or non-homogeneous shape. When a synergistic effect is observed, the number of compounds to use is less than expected, and this turns to dose reduction, a goal that should be pursued when considering the clinical management of human diseases (i.e., the reduction of adverse effects and of costs for expensive drugs).

## 4. Mechanical Bases of Synergistic Effects

From a scientific point of view, it is not sufficient to know that a synergistic effect exists. Pharmacological and clinical research has led to the conception that at least four mechanisms may be involved.

### 4.1. Synergistic Multi-Target Effects

A compound, a mixture or a plant extract may act not only, as expected, on a single target, but may impact on different targets; i.e., all functional or structural cell constituents, such as metabolites, receptors, enzymes, ion channels, transporters, nucleic acids, ribosomes, and proteins [16].

### 4.2. Modulation of Pharmacokinetic or Physicochemical Effects

Synergistic effects may also impact on the physicochemical properties, including solubility, of a compound or mixture, providing an improvement of the so-called bioavailability [17].

### 4.3. Interference with Resistance Mechanisms

Synergistic effects may be observed on drug-resistant microorganisms (bacteria, fungi) or cancer cells, thanks to the presence of natural derivatives that may antagonize the development of drug resistance (supplied together with antibiotics or cancer drugs) [18].

### 4.4. Elimination or Neutralization Potential

A compound or a plant extract may have the ability to remove or neutralize the toxic effect of a drug (synthetic or not), even reducing or nullifying its adverse effects and resulting in treatment amelioration.

With no comprehensive claims, the possible mechanisms of synergistic effects are summarized in Table 1. However, it is of note that the most commonly reported mechanism in the literature is synergistic multi-target effects.

## 5. An Introduction to Generic Plant–Drug Interactions

### 5.1. The Social Problem

Drug interaction is a well-known clinical issue. With the growing number of newly discovered diseases (in the last 50 years), the simultaneous elongation of life expectancy and the detection capacity of new medical technologies has increased the number of prescribed therapies, and this problem should not be undervalued. Besides, if we consider drug interaction as the interaction between pharmacological substances and herbal products or plant extracts (for example, self-prescribed natural products), one could argue that only the tip of the iceberg is emerging, although a huge hidden problem exists. In a larger perspective, it has been estimated that more than 80% of people in Africa and Asia and 60–70% of the American population are using herbal medicines, together with an increasing number in developed countries [2]. Moreover, the amount of related business is increasing day after day. According to the 2007 National Health Interview Survey (NHIS), US adults spent $33.9 billion out of pocket on CAM (in the previous year), and this amount was equivalent to 1.5% of total US health care expenditure and 11.2% of out-of-pocket spending in 2007 [34]. Therefore, the use of plant extracts or natural compounds across the entire world seems to be expected to increase in the next few years and, consequently, the risk of natural products–drug interaction remains challenging.

A critical approach to herbal–drug interactions, adverse effects, and clinical efficacy should be always borne in mind when recommending or taking botanical products, with special attention on serious adverse events [35]. Sometimes, professionals forget that a substance being natural does not mean that it is automatically harmless or innocuous (as many non-professionals believe). Another point to consider is the high variability in plant–drug interactions, due to the distinct methods of preparation or natural product extraction techniques, which are often not standardized and in some countries are not under appropriate regulatory controls.

Marked plant–drug interactions are derived from physiological and pathological individual differences, such as inter-individual variances in drug metabolism, age, gender and the presence of co-morbidities in elderly patients [36]. It is worth mentioning that it is not easy to discern when a patient uses a conventional drug together with a plant-derived compound. Only through a patient interview is some necessary information accessible, as reported by different works [37,38].

### 5.2. Most Common Examples of Detrimental Interactions

The vast majority of natural products and drugs are administered orally, which is a comfortable and easy way to administer a substance. The efficacy of compounds relies on oral bioavailability, which in turn depends on formulation design, biochemical properties, and gastrointestinal system pathophysiology [39]. In this context, the preclinical model Caco-2 cells have been investigated in plant–drug interactions, considering drug bioavailability [40]. Caco-2 cells can predict the probable gastrointestinal permeability of drugs. Indeed, they express cytochrome P450 enzymes and transporters typical of human small intestine enterocytes. For these reasons, Caco-2 cells represent a good screening tool in plant–drug interactions, where they are becoming more attractive [41]. Furthermore, an interesting attempt has been made regarding their ability to inhibit the membrane solute carrier (SLC) family, that together with ATP-binding cassette (ABC) transporters makes up the plasma membrane transporters [42]. The medicinal plant *Salvia miltiorrhiza* L., which consists of lithospermic acid (LSA), rosmarinic acid (RMA), salvianolic acid A (SAA), salvianolic acid B (SAB), and tanshinol (TSL), components of the herbal medicine Danshen, has been assessed regarding the function of different organic anion transporters (OATs) that belong to SLC transporters. The authors demonstrated a strong interaction potential for RMA and TSL, where both act on hOAT1 and hOAT3: in addition the authors showed that LSA could act on hOAT3 [43]. Similar results were obtained in a recent work, in which the organic anion transporting polypeptides (OATP) family was investigated (OATP derives from the SLC family and may transport organic anions through the plasma membrane). The researchers found that the flavonoids apigenin, kaempferol, and quercetin may inhibit OATP1A2 and OATP2B1-mediated drug uptake (i.e., atorvastatin and fexofenadine) in HEK293 cells [44], and as flavonoids may reach high concentrations in the gastrointestinal tract, a real plant–drug interaction may exist.

St John’s wort (*Hypericum perforatum*) is one of the most common natural remedies for depression. It has been demonstrated that St John’s wort is a potent inducer of CYP3A4 and P-glycoprotein (P-gp), influencing the blood concentrations of several drugs, such as amitriptyline, cyclosporine, digoxin, fexofenadine, indinavir, methadone, midazolam, nevirapine, phenprocoumon, simvastatin, and theophylline, whereas it did not alter the pharmacokinetics of carbamazepine, dextromethorphan, mycophenolic acid, and pravastatin. Many other relevant clinical interactions have been observed, such as with warfarin, verapamil, tacrolimus, and so on (for a comprehensive review, see [45]). Also, Ginkgo plant has been intensively studied; several reports showed effects of bleeding after assumption of *Gingko biloba* extract with simultaneous use of aspirin or warfarin [46]. In another case, *Ginkgo biloba* negatively influenced the effect of the antiretroviral drug, Efavirenz (EFV), in an HIV-infected male patient, probably interacting with the EFV metabolism enzymes CYP2B6 and CYP3A4 [47]. Grapefruit juice is another botanical extract that may affect the bioavailability of different drugs ingested with it, such as statins, antihypertensives, and antiretrovirals. This juice causes important drug interactions through CYP3A inhibition, to the extent that the Food and Drug Administration (FDA) warned physicians about the potential and demonstrated interaction of drugs with grapefruit [48].

### 5.3. Most Common Examples of Beneficial Interactions

Not all plant–drug interactions are detrimental. Auspiciously, concrete cases may demonstrate a positive interaction between a plant or herbal product and a conventional drug, which results in a potential increase in drug efficacy or in a possible reduction of adverse effects. For example, the Chinese medicinal plant *Tripterygium wilfordii* has been shown to have immunosuppressive activity by prolonging heart and kidney allograft survival, exhibiting synergy with cyclosporine in a rat model [49]. Moreover, *Acacia confusa* bark extract and its active compound gallic acid were used in a rat model of liver injury, and demonstrated hepatoprotective effects due to antioxidant enzyme modulation, lipid peroxidation inhibition and CYP2E1 activation [50]. Positive effects were also observed in rats co-administrated with herbal medicines and metronidazole. The authors used natural antimalarial products isolated from *Nauclea lafifolia* (Nifadin, Niprisan, and Niprd/92/001/1-1) and found an increase in the serum concentration of metronidazole, thus enhancing the antibiotic potential [51]. In a rat model, garlic (250 mg/kg) combined with captopril demonstrated a synergistic interaction in diminishing blood pressure and inducing angiotensin-converting enzyme (ACE) inhibition [52]. Moreover, a 3-month randomized, double-blind, clinical trial was conducted in beta-thalassemia major patients receiving silymarin, a flavonolignan complex isolated from *Silybum marianum*, in combination with conventional desferrioxamine therapy [53]. The authors showed beneficial results on thalassemia, especially in the treatment of iron-load effects. Another study investigated the concomitant administration of three first-line antiretroviral drugs (lamivudine, stavudine, and nevirapine) and a plant *Andrographis paniculata,* used as an immunostimulant. The researchers showed a beneficial anti-anorexia effect, as well as the modulation of hematological and biochemical indices, with erythrocytes and leucocytes increasing without an associated increase in cholesterol and high-density lipoproteins levels [54].

### 5.4. Controversial and Incomplete Data on Plant–Drug Interactions

Alongside detrimental and beneficial interactions, the vast majority of plant–drug interactions fall into this category, whereas the incomplete data represent the issue we need to face and solve in the near future. Besides this, almost all previously cited works are based on preclinical studies, so that caution should be taken when considering the administration or self-administration of natural products in humans. Consequently, it is crucial to distinguish basic research, with no patient inclusion in which all incomplete data interactions fall, from a clinical trial or meta-analysis, where the results should be regarded as more reliable. In this last case, many examples exist and are frequently associated with detrimental interactions (for a comprehensive review, see [35]). However, controversial results do exist: an example is a ginger, the rhizome of *Zingiber officinale*, a plant traditionally used in the digestive system for its antispasmodic and anti-nausea properties [55]. It has been shown that ginger may interfere with the anticoagulant warfarin, as well as ginkgo, but a strong demonstration is still lacking, and some works have reported that ginkgo and ginger (at recommended doses) did not affect clotting status [56,57].

These short reports about beneficial, detrimental, controversial and incomplete data regarding plant–drug interactions are not exhaustive, and they are outside of the scope of this review. They are merely included to show the extent to which the picture is still complex. For a more detailed discussion, see the works of Liu*,* et al. [58], and Izzo [59].

## 6. Plant–Drug Interaction: Classical Chemotherapeutic Agents and Plant Derivatives

A cancer cell is broadly conceived as a biological entity characterized by continuous uncontrolled growth and abnormal proliferation. This kind of cell may generate a tumor, an irregular mass of tissue, which may be solid or not, that can be differentiated as malignant or benign forms [60]. While the benign form is usually non-progressive and not harmful, malignant entities may grow quickly, invade and metastasize, and potentially result in death. Indeed, cancer is the second-leading cause of death in the USA and developed countries, pushing many researchers and physicians to work hard to fight against cancer [61]. In this context, it is important to use multiple strategies to achieve clinical success, to impact on human survival and to ameliorate life quality. One option is phytomedicine, which can potentially have beneficial effects when properly associated with conventional drugs, both as a preventive approach or targeted-oriented strategy.

### 6.1. The Synergy between Classical Chemotherapeutic Agents and Plant Derivatives

As mentioned above, the synergistic effects between plant-derived bioactives and conventional chemotherapeutic agents may lie in their resistance mechanisms, namely the ability of natural derivatives to antagonize drug resistance or to enhance drug properties, or possibly in the mitigation of side effects.

Several plant-derived products have been studied in association with conventional drugs to find a synergistic effect, and this review attempts to summarize the progress reported to date. It is of note that 20–30% of frequently used chemotherapeutics derive from plants [62].

We may subdivide literature works into three partially overlapping categories: essential oil derivatives, polyphenol derivatives, and a more general class of biochemical plant derivatives. This review focuses on the synergistic combination of the most often used classical chemotherapeutic agents, such as paclitaxel, docetaxel, irinotecan, topotecan, vinblastine, doxorubicin, 5-fluorouracil, mitomycin C and cisplatin with plant-derived bioactives. Synergistic combination requires the use of two compounds together, i.e., association in the same experiment (mixing) of a chemotherapeutic agent and plant derivatives, and not simply the comparison or the use of substances in parallel.

#### 6.1.1. Essential Oil Derivatives

Essential oils are complex mixtures of molecules that belong to two different biosynthetic families, phenylpropanoids and terpenoids, but only the latter (which is volatile) constitutes the main components of essential oils [63]. Different works have shown the significant anticancer effects of plant products in combination with chemotherapy agents. This review analyzed the most common and purchasable essential oils used worldwide and found in the literature, such as geraniol, β-elemene, eugenol, β-caryophyllene, d-limonene, and thymoquinone.

In 2011, geraniol was reported to be effective in diminishing tumor mass volume (70% compared to control), acting on cell cycle and apoptosis pathways and enhancing docetaxel chemosensitivity. This essential oil was associated with docetaxel or taxotere in a xenograft mouse model of PC-3 prostate cancer cells [64]. The platinum-resistant ovarian cancer cell line A2780/CP70 cells were treated with β-elemene, a sesquiterpene derived from a variety of plants, plus a taxane, and resulted in a striking reduction in cell viability and increased cell apoptosis [65]. Moreover, β-elemene markedly inhibited cell growth and proliferation and increased cisplatin cytotoxicity in human bladder cancer 5637 and T-24 cells. It also enhanced cisplatin sensitivity and amplified cisplatin cytotoxicity in in vitro cell models (such as brain, cervix, breast, colorectal, ovary, and small-cell lung cancer) [65,66]. Eugenol is an essential oil extracted from clove, nutmeg, cinnamon, basil, and bay leaf, while sulforaphane is a compound obtained from cruciferous vegetables, such as broccoli, Brussels sprouts or cabbages. Both were tested together with gemcitabine in HeLa cervical cancer cells and showed no significant cell death increase, indicating that cell cytotoxicity was proportional to gemcitabine alone [67]. Moreover, cisplatin combined with methyl eugenol significantly enhanced the anticancer activity of the drug against HeLa cells, acting on cell apoptosis induction, caspase-3 activity, cell cycle arrest and mitochondrial membrane potential loss [68]. β-caryophyllene, a sesquiterpene present in essential oils of various plants, and paclitaxel were tested against MCF-7 (human breast cells), DLD-1 (human colon cells) and L-929 (mouse fibroblasts) tumor cell lines. β-caryophyllene potentiated the anticancer activity of paclitaxel on those cell lines, probably facilitating the passage of paclitaxel through the plasma membrane [69]. β-caryophyllene oxide was also combined with doxorubicin in 2 ovarian cancer cell lines (sensitive A2780 and partly resistant SKOV3) and 2 lymphoblast cancer cell lines (sensitive CCRF/CEM and completely resistant CEM/ADR) [70]. The authors claimed that doxorubicin acted synergistically with β-caryophyllene oxide in SKOV3 and CCRF/CEM cells, while no effect was appreciable on CEM/ADR resistant cells. Similar to the previous work, β-caryophyllene and β-caryophyllene oxide were evaluated for their potential chemosensitizing properties in Caco-2, CCRF/CEM, and CEM/ADR5000 cancer cell lines in association with doxorubicin [70]. The study reported that both sesquiterpenes interfered with the ABC pump function, enhancing doxorubicin cytotoxicity by increasing its intracellular accumulation and oxidative stress status. Moreover, d-limonene, a cyclic terpene obtained commercially from citrus fruits, was used in human prostate carcinoma DU-145 and normal prostate epithelial PZ-HPV-7 cells in combination with docetaxel. The synergistic effects resulted in higher reactive oxygen species (ROS) generation, glutathione depletion and in caspase activity increase [71]. Thymoquinone, a constituent of the *Nigella sativa* essential oil, has been studied in its association with doxorubicin on human cells of HL-60 leukemia, 518A2 melanoma, HT-29 colon, KB-V1 cervix, and MCF-7 breast carcinomas. Thymoquinone improved the anti-neoplastic properties of doxorubicin by inhibiting cancer cell growth [72]. Thymoquinone was also combined with paclitaxel to study the genetic networks involved in their actions using pathway-focused real-time protein chain reaction (PCR) panels, which investigated apoptosis, cell proliferation, growth factor activity and cytokine activity [73]. The work proved that in triple-negative breast cancer cells, thymoquinone caused cytotoxicity and apoptosis increase, and wound healing inhibition, besides sensitizing cancer cells to paclitaxel through extrinsic apoptosis, tumor suppressor genes, and p53 signaling. The synergistic effects were demonstrated in mice breast cancer models, as cancer cell growth decreased. Besides, thymoquinone was associated with docetaxel in DU-145 hormone- and drug-refractory prostate cancer cells, inducing significant synergistic cytotoxicity and apoptosis through the PI3K/Akt signaling pathway [74]. Similarly, thymoquinone enhanced cisplatin- and docetaxel-induced cytotoxicity in 2 triple-negative breast cancer cell lines (with mutant p53) [75], again underlining the synergistic effects of thymoquinone. Two independent works of the same group used irinotecan (CPT-11)-resistant LoVo colon cancer cells to study the effects of thymoquinone, and both demonstrated as this phytochemical can modulate JNK (c-Jun N-terminal kinase) and p38 pathways affecting ERK1/2, PI3K or mitochondrial outer membrane permeability and the autophagy process [76,77]. Also, topotecan was used in combination with thymoquinone in acute myelogenous leukemia cells. The combination regimen (which was better with a pre-exposure) reduced cell proliferation with an increase in Bax/Bcl2, p53, and caspase-3 and -9 expression levels [78]. Similarly, in human colorectal cancer cells, thymoquinone synergistically increased topotecan effectiveness with p53- and Bax/Bcl2-independent mechanisms, by decreasing proliferation and lowering cytotoxicity [79].

#### 6.1.2. Polyphenol Derivatives

Polyphenols are natural substances characterized by the presence of multiple phenol structural units with the ability to modulate oxidative stress in cancer cells [80]. A wide variety of studies have investigated the anticancer effects of plant products in combination with conventional chemotherapy drugs. This review focused on the most common polyphenols reported in the literature, such as resveratrol, genistein, curcumin and quercetin.

In cell models of acute myeloid leukemia, resveratrol (a stilbene that is present in grapes, wine, peanuts, soy, and many other products) [81] induced cell growth arrest and apoptotic death in doxorubicin-resistant AML (Acute Myeloid Leukemia) cells and downregulated MRP1 (*ABCC1* gene codifying for ABC transporter) expression. Three doxorubicin-resistant AML cell lines (AML-2/DX30, AML-2/DX100, AML-2/DX300) were usedk suggesting that resveratrol could overcome doxorubicin resistance or sensitize doxorubicin-resistant AML cells to anti-leukemic agents [82]. Furthermore, resveratrol inhibited the proliferation of multiple human myeloma cell lines and potentiated the apoptotic effects of bortezomib and thalidomide [83,84].

Genistein (a phytoestrogen occurring in *Glycine soja* seeds) was tested in numerous works [85]. Its synergistic combination with cisplatin was evaluated with camptothecin in two human cancer cell lines (HeLa and OAW-42) [86,87], with gemcitabine in two murine xenografts of human pancreatic carcinoma cells (COLO 357 and L3.6pl) [88], and with hydroxycamptothecin in murine xenografts of human bladder carcinoma cells (TCC-SUP) [89]. Moreover, the micronutrient mineral selenium (Se) and genistein together with conventional drugs cisplatin and mitomycin C were used in human peripheral lymphocytes used against cytotoxic agent-induced apoptosis, underlining the protective role of Se and genistein in blood cells [90].

Curcumin, derived from *Curcuma longa* (turmeric) rhizomes, has a long history of use, and its anticancer potential [91,92] was investigated in human (T98G, U87MG, and T67) and rat (C6) glioma cell lines. It was demonstrated that curcumin was able to suppress growth and chemoresistance induced by chemotherapeutic agents (cisplatin, etoposide, camptothecin, and doxorubicin) and radiation [93]. Also, in human ovarian carcinoma cells, curcumin associated with cisplatin or oxaliplatin increased the resistant ovarian cancer cells’ sensitivity to drugs, both in wild-type and in cisplatin-resistant cells, eliciting a reduction in cell cycling and increased apoptosis [94]. Besides this, curcumin inhibited human colon cancer cell line HT-29 growth, when synergically treated with 5-fluorouracil [95]. In MDA-MB-231 cells (highly metastatic breast cancer cells), curcumin associated with ixabepilone, cisplatin, vinorelbine, or everolimus showed cell cycle arrest, a decrease in cell viability and apoptosis induction [96]. Paclitaxel-resistant breast cancer cells and a human breast cancer xenograft model were used to test the synergistic effects of curcumin and paclitaxel. The results indicated that curcumin had a therapeutic action in preventing breast cancer metastasis in a preclinical model by suppressing nuclear factor kappa B (NF-kB) [97].

Quercetin, a flavonol with multiple benefits found in many plants, fruits and vegetables [98], was shown to exhibit synergistic effects with cisplatin in human malignant mesothelioma cell line, with doxorubicin in neuroblastoma and Ewing’s sarcoma cell lines, with temozolomide in human astrocytoma cell line, and with doxorubicin in an established breast cancer in mice [99,100,101,102].

### 6.2. Biochemical Plant Derivatives

Aside from essential oil and polyphenol derivatives, many other herbal compounds or complex mixtures may exert synergistic anticancer properties associated with conventional chemotherapeutic drugs. For example, l-canavanine, an antimetabolite found in several plants of the *Fabaceae* family, which is hardly toxic alone, potentiated the cytotoxicity of vinblastine and paclitaxel in 2 cell models (HeLa and hepatocellular carcinoma cells) [103]. Moreover, in an orthotopic pancreatic cancer mouse model with PANC-1 cells, β-carboline (an alkaloid from the plant *Rauwolfia vomitoria*)-enriched extract in combination with gemcitabine reduced tumor burden and metastatic potential in gemcitabine non-responsive tumor [104]. The same plant, *Rauwolfia vomitoria*, in combination with carboplatin, was able to increase chemosensitivity in ovarian cancer cells (OVCAR-5, OVCAR-8, SHIN-3) and to inhibit tumor growth in a mouse model with intraperitoneal metastasis and massive ascites formation [105]. *Garcinia benzophenones* (obtained from *Garcinia* species) was tested on HT29 colon cancer cells in association with different chemopreventive agents and demonstrated a great ability to block cancer cell growth [106]. Another work showed that red beetroot (*Beta vulgaris*) extract together with doxorubicin induced synergistic antiproliferative effects against pancreatic (PaCa), breast (MCF-7) and prostate (PC-3) tumor cells [107].

## 7. Concluding Remarks

Phytomedicine and ethnopharmacology have increased in popularity in the last decades, especially when considering their application in the vast field of human diseases [108,109,110,111,112]. The results of the above-reported literature studies suggest that plant-derived compounds have a high impact as therapeutic agents, both alone or in combination with conventional drugs. Thus, evidence-based phytotherapy should be considered a valid option when treating human disorders, not only for low or mild-grade diseases but also for more complex and problematic diseases, such as cancer. Certainly, the pre-clinical data shown in this review need to be confirmed by robust clinical trials (randomized double-blind), and along this line of scientific research, a good number of clinical trials has been reported. Indeed, bibliographic research using “cancer and herbal” and “cancer and plant” as keywords retrieved 78 and 149 studies on the website of ClinicalTrials.gov (December 2018), even if not strictly related to synergy [113]. This gives us an idea of the undoubtedly increasing interest for CAM and should bring new attention to this field of research, which is sometimes neglected by the main financing channels. Also, it should be kept in mind that the growing use of herbal products, either self-prescribed or integrated with conventional drugs, will eventually contribute to intensifying the incidence of plant–drug interactions. Thus, every health practitioner should be aware of this important issue. However, considering cancer patients, there is the obvious need to invest in more clinical work exploring the administration of herbal remedies, focusing not only on avoiding harmful interactions but also on providing valid support if a scientific synergistic demonstration guarantees effectiveness. Indubitably, an accessible, worldwide, free database of herb–drug interactions is not only desirable but is now becoming a concrete need for clinicians; a need not strictly related to cancer treatment.

## Figures and Tables

**Figure 1 medicina-55-00110-f001:**
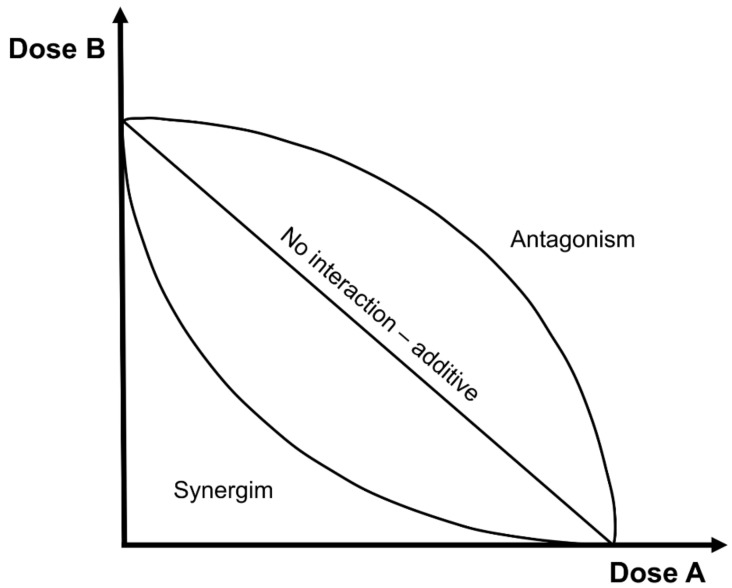
Representation of the isobole method.

**Table 1 medicina-55-00110-t001:** Literature examples of the four mechanisms involved in synergistic effects.

Mechanism	Plants Involved	References
Synergistic multi-target effects	Herbal pair Chuanxiong rhizome and *Paeonia albifora*	[19]
	*Ocimum sanctum* flavonoid vicenin-2	[20]
	Cannabis extract *delta*9-*trans*-tetrahydrocannabinol	[21]
	St. John’s wort (*Hypericum perforatum*)	[22]
Pharmacokinetic or physicochemical effect modulations	Yin-Chen-Hao-Tang (YCHT), a Chinese herbal formula (*Herba artemisiae Yinchenhao + fructus gardeniae gasminoidis + radix et rhizoma rhei*)	[23]
	*Ammi visnaga* aqueous extract	[24]
	*Hypericum perforatum* flavonoids	[17]
	Grapefruit juice (*Citrus × paradise*)	[25]
	*Panax ginseng*	[26]
Interference with resistance mechanisms	Seven commercially available terpenoids	[27]
	Three commercially available flavonoids (apigenin, quercetin, naringenin)	[28]
	*Pelargonium graveolens* essential oil	[29]
	Nine herbal extracts and 23 isoflavonoids	[30]
Elimination or neutralization potential	*Ocimum basilicum* constituent nevadensin	[31]
	PHY906, a mixture of *Scutellaria baicalensis*, *Glycyrrhiza uralensis*, *Paeonia lactiflora*, *Ziziphus jujube*	[32]
	*Silybum marianum* (Silymarin) and *Glycyrrhiza glabra* (Glycyrrhizin) extracts	[33]

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
