# Peer review of "Synergistic Effects of Plant Derivatives and Conventional Chemotherapeutic Agents: An Update on the Cancer Perspective"

_medicina, 2019, doi:10.3390/medicina55040110_

Round 1

Reviewer 1 Report

This is a review article written by Pezzani et al. to describe the synergistic effects of plant derivatives and conventional chemoagents on cancer treatment. This review gives a clear definition of “synergy” and introduces several models for evaluating synergistic effects. Moreover, this article makes a comprehensive review on the synergistic combination of three categories of plant derivatives and chemotherapeutic agents, and describes them in an appropriate format. I just have one minor opinion: I find the paper of reference #86 has been retracted. I recommend that this article is not suitable to be included in this review.

Author Response

This is a review article written by Pezzani et al. to describe the synergistic effects of plant derivatives and conventional chemoagents on cancer treatment. This review gives a clear definition of “synergy” and introduces several models for evaluating synergistic effects. Moreover, this article makes a comprehensive review on the synergistic combination of three categories of plant derivatives and chemotherapeutic agents, and describes them in an appropriate format. I just have one minor opinion: I find the paper of reference #86 has been retracted. I recommend that this article is not suitable to be included in this review.

Answer: Thank you the reviewer for the encouraging comments. As suggested by reviewer, we deleted the reference n° 86 and its related description in the manuscript.

Reviewer 2 Report

This is a relatively useful review. It states more about general plant derivatives in combination with chemotherapy. The search terms could poses a problem. Many literature only put the Latin name of the plant without putting "herb (s)", "plant (s)", "phyto*", in the article, which are the searching terms of this review. Therefore, there is a potential that many related articles could be missing.

another issue is phytochemicals. Many algae derivatives are considered phytochemicals while the title suggest only plant origin of phytochemicals are reviewed. This is a bit narrow. Better to include algal derivatives including fucoidan, fucoxanthin, astaxanthin etc.

Author Response

This is a relatively useful review. It states more about general plant derivatives in combination with chemotherapy. The search terms could poses a problem. Many literature only put the Latin name of the plant without putting "herb (s)", "plant (s)", "phyto*", in the article, which are the searching terms of this review. Therefore, there is a potential that many related articles could be missing.

Answer: The reviewer is correct, some authors or articles refer only to plant Latin name, however it is impossible to do a literature review searching for every single plant Latin name. Morevoer, when we prepared this review we tried to search the term “plant(s)”, “herb(s)”, “phyto” not only in the Title, but also in the Abstract. Of course, not always in the Abstract such terms can be found, nonetheless this kind of search can guarantee best results if compared to title alone.

Another issue is phytochemicals. Many algae derivatives are considered phytochemicals while the title suggest only plant origin of phytochemicals are reviewed. This is a bit narrow. Better to include algal derivatives including fucoidan, fucoxanthin, astaxanthin etc.

Answer: After a literature research we decided to avoid to extend our research also to algae compounds, as this will require another work. We added a sentence at the end of introduction to remark this: “Algae derivatives or extracts were intentionally excluded, as their investigation would require a dedicated review.